# Psychological distress among healthcare providers during COVID-19 in Asia: Systematic review and meta-analysis

Siew Mooi Ching[1,2,3,4], Kar Yean Ng[1], Kai Wei Lee[5,6], Anne Yee[7], Poh Ying Lim[8], Hisham Ranita[9], Navin Kumar Devaraj[1,2], Pei Boon Ooi[3], Ai Theng Cheong[1]*

1 Department of Family Medicine, Faculty of Medicine and Health Sciences, Universiti Putra Malaysia, Serdang Selangor, Malaysia, 2 Malaysian Research Institute on Ageing, Universiti Putra Malaysia, Serdang, Selangor, Malaysia, 3 Department of Medical Sciences, School of Medical and Life Sciences, Sunway University, Bandar Sunway, Selangor, Malaysia, 4 Centre for Research, Bharath Institute of Higher Education and Research, Selaiyur, Chennai, Tamil Nadu, India, 5 Department of Pre-Clinical Sciences, Faculty of Medicine and Health Sciences, Universiti Tunku Abdul Rahman, Kajang, Malaysia, 6 Centre for Research on Communicable Diseases, Universiti Tunku Abdul Rahman, Kajang, Malaysia, 7 Department of Psychological Medicine, Faculty of Medicine, University Malaya, Kuala Lumpur, Malaysia, 8 Department of Community Health, Faculty of Medicine and Health Sciences, Universiti Putra Malaysia, Serdang, Selangor, Malaysia, 9 Universiti Malaya Library, Universiti Malaya, Kuala Lumpur, Malaysia

* cheaitheng@upm.edu.my

**Data Availability Statement:** All relevant data are within the paper and its Supporting information files.

## Abstract

### Introduction

COVID-19 pandemic is having a devastating effect on the mental health and wellbeing of healthcare providers (HCPs) globally. This review is aimed at determining the prevalence of depression, anxiety, stress, fear, burnout and resilience and its associated factors among HCPs in Asia during the COVID-19 pandemic.

### Material and methods

We performed literature search using 4 databases from Medline, Cinahl, PubMed and Scopus from inception up to March 15, 2021 and selected relevant cross-sectional studies. Publication bias was assessed using funnel plot. Random effects model was used to estimate the pooled prevalence while risk factors were reported in odds ratio (OR) with 95% CI.

### Results

We included 148 studies with 159,194 HCPs and the pooled prevalence for depression was 37.5% (95%CI: 33.8–41.3), anxiety 39.7(95%CI: 34.3–45.1), stress 36.4% (95%CI: 23.2–49.7), fear 71.3% (95%CI: 54.6–88.0), burnout 68.3% (95%CI: 54.0–82.5), and low resilience was 16.1% (95%CI: 12.8–19.4), respectively. The heterogeneity was high (I2>99.4%). Meta-analysis reported that both females (OR = 1.48; 95% CI = 1.30–1.68) and nurses (OR = 1.21; 95%CI = 1.02–1.45) were at increased risk of having depression and anxiety [(Female: OR = 1.66; 95% CI = 1.49–1.85), (Nurse: OR = 1.36; 95%CI = 1.16–1.58)]. Females were at increased risk of getting stress (OR = 1.59; 95%CI = 1.28–1.97).

**Funding:** The author(s) received no specific funding for this work.

**Competing interests:** The authors have declared that no competing interests exist.

**Abbreviations:** Asi-3, Anxiety Sensitivity Index-3; BAS, Beck Anxiety Scales; BDI, Beck Depression Inventory; CAS, Coronavirus Anxiety Scale; CCMD-3, Chinese Classification and the Diagnose Criterion of Mental Disorder; CD-RISC, Connor-Davidson Resilience Scale; CES-D, Center for Epidemiology Studies-Depression; COVID-19, Novel coronavirus 2019; DASS-21, Depression-Anxiety-Stress Scale-21; FCV-19S, Fear of Coronavirus 2019 Scale; FS-HPs, Fear Scale for Healthcare Professionals; GAD, Generalized Anxiety Disorder; HADS, Hospital Anxiety and Depression Scale 14 items; HAMA, Hamilton Anxiety Rating Scale; HAMD, Hamilton Depression Rating Scale; IUS-12, Intolerance of Uncertainty Scale; MBI, Maslach Burnout Inventory; MBI-HSS, Maslach Burnout Inventory-Human Service Survey; MINI, Mini International Neuropsychiatric Interview; NA, Not available; NRS, Numerical rating scale; OLBI, Oldenburg Burnout Inventory; PHQ, Patient Health Questionnaire; PRISMA, Preferred Reporting Items for Systematic Review and Meta-analyses; PSS, Perceived Stress Scale; SAQ, Safety Attitudes Questionnaire; SAS, Zung's Self-Rating Anxiety Scale; SASRQ, Stanford Acute Stress Reaction Questionnaire; SDS, Zung's Self-rating Depression Scale; STAI, State and Trait Anxiety Inventory; STROBE, Strengthening the Reporting of Observational Studies in Epidemiology; WHO-5, WHO Well-being Index.

## Conclusion

In conclusion, one third of HCPs suffered from depression, anxiety and stress and more than two third of HCPs suffered from fear and burnout during the COVID-19 pandemic in Asia.

## Introduction

The COVID-19 pandemic is both a worldwide healthcare crisis and financial disaster at the biggest scale that has emerged in the last century. With the emergence of new COVID-19 virus variants, COVID-19 is even more infectious and can spread more rapidly through various inter-continentals [1, 2]. To date, about 171 million individuals had contracted the infection caused by this novel coronavirus and more than 3.5 million have succumbed to this virus across 222 countries [3].

There has been increasing concern that COVID-19 has infected nearly 570,000 healthcare providers (HCP) and killed more than 2500 of them in the Americas alone [4]. In the Asia Pacific region, a total of 12,454 HCP have been infected with the novel coronavirus while 171 have succumbed to this virus as of June 11, 2020 [5].

In performing their duties of arresting the spread of COVID-19, the HCP are risking their lives due to a higher risk of virus exposure, high workload demand, irregular or long working hours and increased psychological distress such as depression, anxiety, stress, occupation burnout, fear, low resilience as well as fatigue [1]. In addition, the HCP were barred from taking leaves and separated from their loved ones for up to weeks or even months. Wearing the full personal protective equipment or gear (PPE) that is very uncomfortable for long hours continuously every day while managing patients diagnosed with COVID-19 is extremely exhausting, particularly that this has become a routine task in their daily work. Literature reported that factors associated with personal-, work-, and patient-related burnout among HCPs were those that had direct involvement in COVID-19 management, underlying medical illness, and receiving inadequate psychological support in the workplace [6]. Those with higher total points in the coping score were significantly associated with reduction in anxiety and depression scores [7]. Other significant factors associated with psychological distress inluding but not limited to thought of resignation and reluctant to work, fear of infecting family members, frequent change in infection prevention and control protocol or guideline, and poor social support [8]. All of the aforementioned factors had been determined as factors that are leaving negative psychological impacts on the healthcare workers in Asian countries [8–10].

HCPs experiencing anxiety and depression have impaired physical and mental health and it may affect their wellbeing and work efficacy. In other word, the psychological consequences may contribute to a poorer quality of life among HCPs and suboptimal performance of delivered care. Experiencing these psychological hardships in the long run, can lower the immunity and put HCPs at a higher risk of being infected [11]. Furthermore, HCPs with pre-existing depression may suffer a higher mortality rate if they are hospitalised with COVID-19 [12]. Thus, it is very important to create the awareness on the degree of psychological distress that are encountered by HCPs amidst the COVID-19 pandemic in order to help reduce the incidence of occupation-related burnout and deaths. This can then serve as a platform for the government and policy makers to allocate funding to promote and provide psychosocial support for HCPs during this COVID-19 pandemic which will ultimately led to a better patient care. Recent systematic review involving 32 studies reported on the prevalence of mental health

among healthcare personnel during COVID-19 in Asia but there was no meta-analysis on pooled prevalence [13]. Another recently published systematic review by Md Mahbub Hossain et al involving 35 studies with 41,402 participants reported the prevalence of anxiety and depression during COVID-19 pandemic in South Asia but not in the Asia region specifically [14]. There was no systematic review and meta-analysis from Asia regarding these topics in which the burden of the psychological impacts could be different in view of the varietis of health care system across this region. Therefore, we are performing this systematic review and meta-analysis to determine the prevalence of depression, anxiety, stress, burnout, fear and low resilient among HCPs as well as its associated factors during the COVID-19 pandemic in Asia.

## Materials and methods

### Protocol

This present study is registered with INPLASY (Number: 202140043). We have also adhered to the Preferred Reporting Items for Systematic Reviews and Meta-Analyses (PRISMA) guideline [15] on conducting and reporting this systematic review and meta-analysis result as stated in Table A1 in S1 Table.

### Literature search

Two authors (HR and CSM) performed literature search based on four databases (Medline, Cinahl, PubMed and Scopus databases) systematically and independently for potential articles published in 2020 to 13th of March, 2021. A combination of search terms that consists of (depression OR anxiety OR stress OR burnout, professional OR fatigue OR fear OR resilience, psychological OR adjustment) AND (healthcare workers OR medical staff OR healthcare professionals OR medical personnel) AND (coronavirus OR SARS-COV-2 OR COVID-19) AND (Asia) were used and is stated in Table A2 in S2 Table.

### Selection criteria

The inclusion criteria for this systematic review were as follow:

1. The study design was cross-sectional with a minimum sample size of 100

2. The study stated the prevalence of depression, anxiety, stress, burnout, fear and resilience among HCPs during COVID-19 pandemic

3. The study evaluated depression, anxiety, stress, burnout, fear and resilience based on validated instrument tools or scales

4. The study involved HCPs from Asian countries

5. The studies must be published in English peer-reviewed journals.

   Studies with the following criteria were excluded:

1. Perspective, opinion, review articles, case reports, short communications paper, no full text study and unpublished data

2. Data reported in continuous or qualitative format

3. Outcomes were not clearly defined by validated tools

4. Depression, anxiety, stress, burnout, fear and resilience were reported as independent data

5. Technical error was present in the reported data

6. After full-text articles have been assessed for eligibility, those outcomes were grouped into category of severities which were different from our operational definition.

### Study selection

We performed the study selection according to the PRISMA guidelines. Studies identified using the search strategies were transferred into Endnote software (version 19) for screening, removing duplication and data extraction. Two authors (CSM and NKY) screened the title and abstract to determine the eligibility of the studies. For those potentially eligible articles, further screening on the full text had been performed to determine the availability of data and whether the articles fulfilled the selection or inclusion criteria (types of studies, participants, setting and outcomes). Any doubt on eligibility was resolved by discussion with a third author (KWL). Finally, those full-text articles which fulfilled all selection criteria were kept for data extraction and subsequent quantitative analysis.

### PICO

The participants were HCPs (doctors, dentists, nurses, nurse assistants, midwives, medical assistants, pharmacists and other allied healthcare workers). Exposure was referred to actively providing healthcare-related services in conditions that are high risk for COVID-19 transmission and there is no comparator for the current systematic review. The main outcome for this review was pooled prevalence of depression, anxiety, depression, burnout, fear and resilience among HCPs.

### Data extraction

We used Microsoft Excel to perform the data extraction and recording. The following data were extracted independently by two authors (SMC and KYN) and recorded as: Author information, publication year, country of the study, mean age or median of participants, study design, sample size, number of HCPs with and without symptoms of depression, anxiety, stress, burnout, fear and resilience, screening tool for assessment of depression, anxiety, stress, burnout, fear and resilience, sociodemographic and any clinical characteristics of the respondents.

### Quality assessment

We used the Strengthening the Reporting of Observational Studies in Epidemiology (STROBE) checklist to perform quality assessment on all the included articles [16]. Two authors (CSM and NKY) individually assessed the study quality, and discrepancies were resolved by discussion with third investigator (LKW). STROBE check list consists of 22 items that assessed 6 components in cross-sectional studies. One point would be rewarded for a positive response for each of the items making the total score ranging from 0 to 22. Each article was graded as 'low risk of bias' if STROBE score ≥14/22; or 'high risk of bias' if the score is <14/22 [16]. The STROBE scoring for this systematic review is presented in Appendix A. Studies were included in the analysis regardless of STROBE score and grades. The summary of the quality grade is reported in Table 1.

### Operational definition

Psychological distress in this systematic review covers depression, anxiety, stress, fear, burnout, and low resilience symptoms. Presence of depression symptom was applied to those who scored points that belonged to mild to severe depression domain categories. Similarly, this

**Table 1. Characteristic of 148 studies.**

| | No. of article | No. of country | Method of screening | Number of positve case | Total population |
|---|---|---|---|---|---|
| All | 148 | 23 | | | 159194 |
| Depression [8, 21–25, 27–31, 33, 34, 36–39, 43–50, 54–58, 60–63, 65–67, 69–74, 76–80, 82, 84, 86, 87, 89, 90, 92, 93, 95, 97, 98, 101, 102, 104–106, 109, 111, 113–115, 118, 122, 125, 127–131, 135, 136, 138, 139, 141–144, 147, 149–152, 156–159, 161–163] | 98 | 20 | DASS-21, PHQ-2, PHQ-4, PHQ-9, HADS, HAMD, SDS, MINI, CES-D, CCMD-3, SCL-90, STAI, BDI, WHO-5 | 37630 | 103628 |
| Anxiety [7–9, 22–25, 28–31, 33–40, 43–54, 56–58, 60–63, 65–67, 69–74, 76–80, 82, 84, 86–93, 97, 98, 101–112, 114–116, 118, 120–133, 135–139, 141–143, 147, 149–152, 154–163] | 117 | 21 | DASS-21, GAD-2, GAD-7, HAMA, SAS, MINI, PHQ-4, CCMD-3, HADS, ASI-3, SCL-90, COVID-19 Anxiety Scale, CAS, Dispositional cancer worry scale, STAI, IUS-12 STAI, BAS | 38284 | 99639 |
| Stress [28, 29, 37, 43, 54, 57, 75, 76, 89, 90, 97–100, 102, 104, 105, 111, 118, 121, 122, 125, 127–131, 136, 138, 139, 141, 142, 148, 150, 151, 153, 158, 159, 162, 163] | 40 | 17 | DASS-21, PSS, SASRQ, SAQ, STAI. | 8599 | 34010 |
| Burnout [6, 32, 41, 85, 140, 142, 158] | 7 | 5 | MBI, ProQOL Scale, MBI-HSS, Adopted Queationnaire, OLBI. | 5426 | 8732 |
| Fear [31, 56, 105, 143, 156] | 5 | 4 | FCV-19S, FS-HPS, CCMD-3, NRS, STAI. | 3460 | 4302 |
| Low Resilience [9, 94] | 2 | 2 | CD-RISC | 78 | 484 |

Abbreviation:

ASI-3: Anxiety Sensitivity Index-3;

BAS: Beck Anxiety Scales;

BDI: Beck Depression Inventory;

CAS: Coronavirus Anxiety Scale;

CCMD-3: Chinese Classification and the Diagnose Criterion of Mental Disorder;

CD-RISC: Connor-Davidson Resilience Scale;

CES-D: Center for Epidemiology Studies-Depression;

DASS-21: Depression-Anxiety-Stress Scale-21;

FCV-19S: Fear of Coronavirus 2019 Scale;

FS-HPs: Fear Scale for Healthcare Professionals;

GAD-2: 2-item Generalized Anxiety Disorder;

GAD-7: 7-item Generalized Anxiety Disorder;

HADS: Hospital Anxiety and Depression Scale 14 items;

HAMA: Hamilton Anxiety Rating Scale;

HAMD: Hamilton Depression Rating Scale;

IUS-12: Intolerance of Uncertainty Scale;

MBI: Maslach Burnout Inventory;

MBI-HSS: Maslach Burnout Inventory- Human Service Survey

MINI: Mini International Neuropsychiatric Interview;

NRS: Numerical rating scale

OLBI: Oldenburg Burnout Inventory;

PHQ-2: 2-item Patient Health Questionnaire;

PHQ-4: Patient Health Questionnaire-4;

PHQ-9: 9-item Patient Health Questionnaire;

PSS: Perceived Stress Scale;

SAQ: Safety Attitudes Questionnaire;

SAS: Zung's Self-Rating Anxiety Scale;

SASRQ: Stanford Acute Stress Reaction Questionnaire;

SDS: Zung's Self-rating Depression Scale;

STAI: State and Trait Anxiety Inventory;

WHO-5: WHO Well-being Index;

applied for anxiety, stress and fear symptoms. Burnout is characterized by those who ranked under moderate to severe categories. Low resilience is defined as those classified under a low score category.

## Data synthesis

We use the *Open Meta Analyst* and StatsDirect to perform the meta-analyses [17, 18]. We used a random-effects model (DerSimonian and Laird Method) to calculate the pooled prevalence from multiple studies [19, 20]. The numerator would be the total number of cases summed up of HCPs with positive symptoms; denominator would be total number of HCPs in that study. Values of numerator and denominators of all studies were used to derive the pooled prevalence of outcome variables. The random-effects model was used so that the findings were generalisable and more representative presuming that the studies were randomly selected from a bigger population [21]. We used $I^2$ statistics index to assess the heterogeneity across the studies. $I^2$ index is categorised as low if $< 25\%$, moderate 25–50%, and high $> 50\%$) [20].

## Subgroup analyses

Subgroup analyses are useful to examine the between-group differences in terms of the prevalence as a possible cause of heterogeneity across studies. The prevalence of depression, anxiety, stress, burnout, fear and resilience among HCPs in Asia was determined by subgrouping the countries. The data were further examined by subgrouping the gender and occupational subtype of HCPs. The risk factors for depression, anxiety, stress, burnout, fear and resilience were reported in odds ratio (OR) with a 95% confidence interval (CI).

## Sensitivity analysis

We performed sensitivity analysis by using leave-one-out meta-analysis to examine how each particular study alters the overall performance of the rest of the studies especially the pooled prevalence estimates and heterogeneity.

## Publication bias

We assessed the potential publication bias by visually checking the Funnel plot followed by conducting the Begg's test and Egger's regression test. In Egger's test, a p-value $< 0.01$ was defined as an indicator for publication bias [22].

# Results

## Search result

Fig 1 shows the PRISMA flow diagram used in this review. A total of 2446 articles were identified from four databases: CINAHL (155), Medline (500), PubMed (746) and Scopus (1045). After removing the duplicate records, we performed screening on 1241 articles. Finally, we included 148 studies for systematic review and meta-analysis after further evaluation of eligibility.

## Description of included studies

Table 1 summarizes the main characteristics of 148 studies. A total of 159,194 healthcare providers from 23 different Asia countries were involved in this systematic review and meta-analysis. Almost half (n = 70) of the studies were conducted in China [8, 11, 23–90], followed by 15 studies in India [10, 91–104], 11 studies in Turkey [105–115], 10 studies in Saudi Arabia

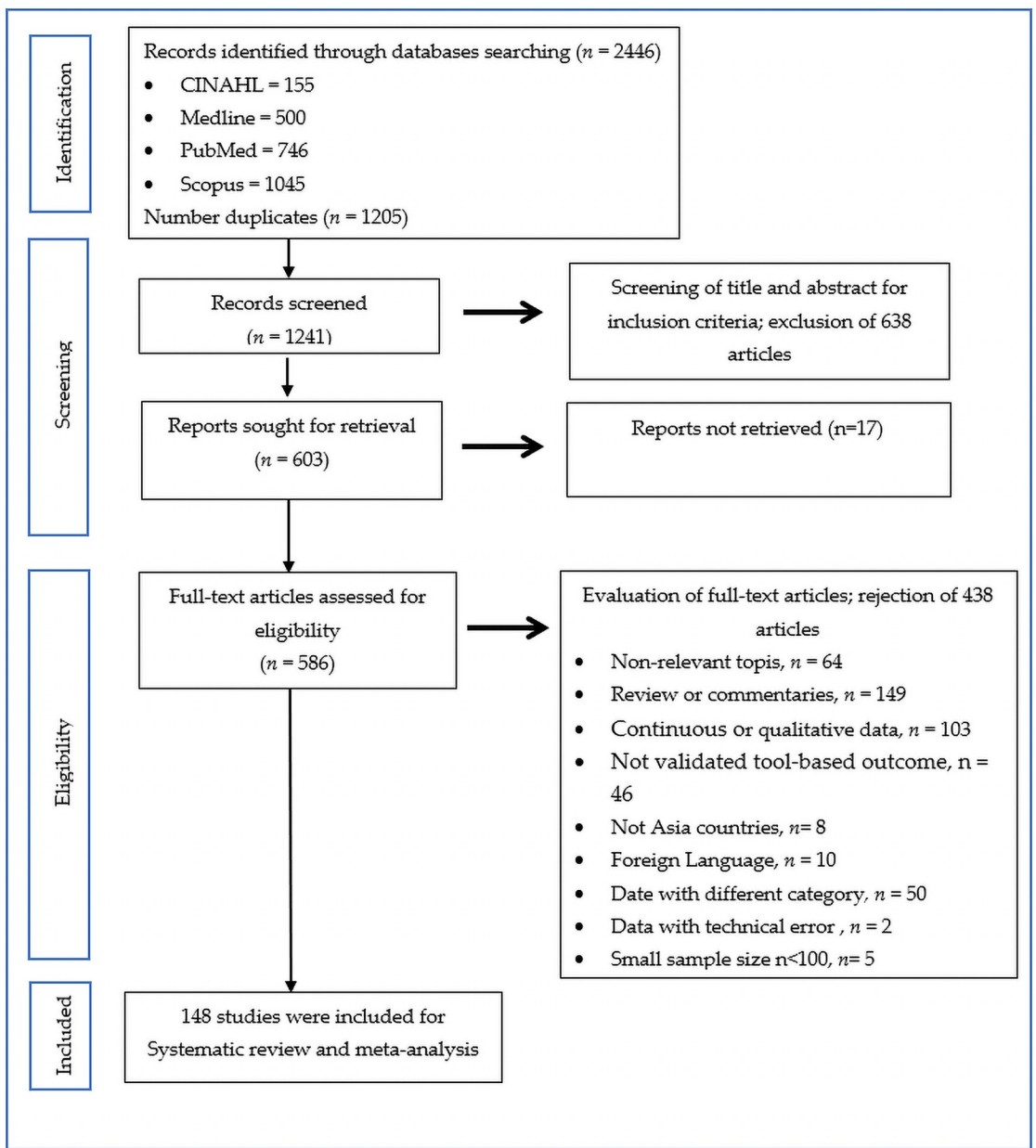

**Fig 1. PRISMA flow diagram of the literature screening process.**

[116–125], 6 studies in Pakistan [126–131], 4 studies in Indonesia [132–135] and Nepal [136–139] respectively, 3 studies in Malaysia [140–142], Singapore [143–145], Japan [146–148] and Iran [149–151] respectively with 2 studies in Oman [152, 153], Jordan [154, 155], Philippines [156, 157] and Bangladesh [158, 159] respectively. Besides, Korea [160], Qatar [161], and Iraq [162] which each had one study, there were also some multinational studies that were conducted, with 2 of them involving several countries in Asian Pacific [9, 163] and another one involving both Egypt and Saudi Arabia [164] (Table A4 in S4 Table).

About two thirds (n = 98) of the studies reported the data on depression [8, 23–27, 29–33, 35, 36, 38–41, 45–52, 56–60, 62–65, 67–69, 71–76, 78–82, 84, 86, 88, 89, 91, 92, 94, 95, 97, 99,

100, 103, 104, 106–108, 111, 113, 115–117, 120, 123, 124, 127, 129–133, 137, 138, 140, 141, 143–146, 149, 151–154, 158–161, 163–165]. Different type of tools or scales were used to diagnosed depression in different papers from different Asian countries (Table 1 and S4 Table), such as 2-item Patient Health Questionnaire (PHQ-2), 9-item Patient Health Questionnaire (PHQ-9), Hamilton Depression Rating Scale (HAMD), Hospital Anxiety and Depression Scale 14 items (HADS), Chinese edition of Zung's Self-Rating Depression Scale (SDS), Center for Epidemiology Studies-Depression (CES-D), Chinese Classification and the Diagnose Criterion of Mental Disorder (CCMD-3), Hospital Anxiety and Depression Scale 14 items (HADS), Symptom Checklist 90, and also Beck Depression Inventory (BDI).

A total of 117 out of 148 studies described the anxiety data in terms of frequency or percentage [8, 10, 11, 24–27, 30–33, 35–42, 45–56, 58–60, 62–65, 67–69, 71–76, 78–82, 84, 86, 88–95, 99, 100, 103–114, 116–118, 120, 122–135, 137–141, 143–145, 149, 151–154, 156–165]. Tools that were used to screen for anxiety were Depression-Anxiety-Stress Scale-21 (DASS-21), 2-item and 7-item Generalized Anxiety Disorder (GAD-2& GAD-7), Hospital Anxiety and Depression Scale 14 items (HADS), Self-Rating Anxiety Scale (SAS), Hamilton Anxiety Scale (HAMA), Patient Health Questionnaire-4 (PHQ-4), Coronavirus Anxiety Scale (CAS), Chinese Classification and the Diagnose Criterion of Mental Disorder (CCMD-3), State and Trait Anxiety Inventory (STAI), Dispositional cancer worry scale, Symptom Checklist 90, and Beck Anxiety Scales (BAS).

There were 40 studies which reported on data regarding stress, using five different type of scales which were Depression-Anxiety-Stress Scale-21 (DASS-21), Safety Attitudes Questionnaire (SAQ), Perceived Stress Scales (PSS), Stanford Acute Stress Reaction Questionnaire (SASRQ), and State and Trait Anxiety Inventory (STAI) [30, 31, 39, 56, 59, 77, 78, 91, 92, 99–102, 104, 106, 107, 113, 120, 123, 124, 127, 129–133, 138, 140, 141, 143, 144, 150, 152, 153, 155, 160, 161, 164, 165].

Only 7 studies investigated data on burnout in term of its' frequency or percentage. The most common tools used to screen burnout among healthcare providers was Maslach Burnout Inventory (MBI), followed by ProQOL Scale of Chinese version, and Oldenburg Burnout Inventory (OLBI) [9, 34, 43, 87, 142, 144, 160]. A validated questionnaire form was adopted from Michelle Post, Public Welfare, Vol. 39, No. 1, 1981, American Public Welfare Association to examine the prevalence of burnout.

On the other hand, 5 studies reported data on fear, using five different scales which were Fear of Coronavirus 2019 Scale (FCV-19S), Fear Scale for Healthcare Professionals (FS-HPs), Chinese Classification and the Diagnose Criterion of Mental Disorder (CCMD-3), Numerical rating scale (NRS), and State and Trait Anxiety Inventory (STAI) [33, 58, 107, 145, 158]. Besides, two studies that reported on low resilience data were using the Connor-Davidson Resilience Scale (CD-RISC) as screening tool [11, 96].

## Pooled prevalence of psychological distress among healthcare providers amidst COVID-19 pandemic

A summary of pooled prevalence of mental illnesses among healthcare providers during COVID-19 pandemic is shown in Table 2 and S1–S5 Figs. The overall pooled prevalence of mild to severe depression among HCPs is 37.5 (95% CI = 33.8–41.3). (Fig 2) Jordan recorded the highest prevalence of depression among HCPs at 78.0% (95% CI = 75.6–80.4), followed by a multicentre study involved Egypt and Saudi Arabia at 69.0% (95% CI = 64.6–73.4), and Iran at 59.6% (95% CI = 37.4–81.9). Pooled prevalence of depression of 52 studies in China was reported as 36.5% (95% CI = 31.7–41.2) while that in Malaysia was 26.6% (95% CI = 17.9–35.3) (S1 Fig).

**Table 2. Summary of pooled prevalence of psychological distress among healthcare providers during COVID-19 pandemic.**

| Domain | Country | N | Total Psychological distress | Total sample size | Prevalence,% (95% CI) | I² (p-value) | Appendix |
|---|---|---|---|---|---|---|---|
| Depression | All | 98 | 37630 | 103628 | 37.5 (33.8–41.3) | 99.49 (<0.001) | Fig 2 |
| | Asian Pacific region | 1 | 51 | 1146 | 4.5 (3.3–5.6) | NA | NA |
| | Bangladesh | 2 | 342 | 782 | 43.5 (33.5–53.4) | 87.95 (0.004) | Fig A1.1 in (S1 Fig) |
| | China | 52 | 22772 | 66052 | 36.5 (31.7–41.2) | 99.51 (<0.001) | Fig A1.2 in (S1 Fig) |
| | Egypt and Saudi | 1 | 294 | 426 | 69.0 (64.6–73.4) | NA | NA |
| | India | 9 | 1804 | 5573 | 33.6 (27.9–39.3) | 94.62 (<0.001) | Fig A1.3 in (S1 Fig) |
| | Indonesia | 2 | 268 | 1326 | 20.2 (14.1–26.4) | 87.70 (0.004) | Fig A1.4 in (S1 Fig) |
| | Iran | 2 | 548 | 928 | 59.6 (37.4–81.9) | 98.10 (<0.001) | Fig A1.5 in (S1 Fig) |
| | Japan | 1 | 237 | 848 | 27.9 (24.9–31.0) | NA | NA |
| | Jordan | 1 | 907 | 1163 | 78.0 (75.6–80.4) | NA | NA |
| | Korea | 1 | 20 | 115 | 17.4 (10.5–24.3) | NA | NA |
| | Malaysia | 2 | 413 | 1449 | 26.6 (17.9–35.3) | 91.99 (<0.001) | Fig A1.6 in (S1 Fig) |
| | Nepal | 2 | 295 | 879 | 33.2 (24.9–41.6) | 86.17 (0.007) | Fig A1.7 in (S1 Fig) |
| | Oman | 2 | 550 | 1541 | 38.6 (25.9–51.3) | 95.19 (<0.001) | Fig A1.8 in (S1 Fig) |
| | Pakistan | 4 | 4897 | 10790 | 44.6 (19.3–69.9) | 99.54 (<0.001) | Fig A1.9 in (S1 Fig) |
| | Qatar | 1 | 54 | 127 | 42.5 (33.9–51.1) | NA | NA |
| | Saudi Arabia | 5 | 1313 | 2483 | 44.2 (28.7–59.6) | 98.20 (<0.001) | Fig A1.10 in (S1 Fig) |
| | Singapore | 2 | 1043 | 3197 | 41.7 (21.5–61.9) | 95.03 (<0.001) | Fig A1.11 in (S1 Fig) |
| | Singapore and India | 1 | 96 | 906 | 10.6 (8.6–12.6) | NA | NA |
| | Turkey | 6 | 1691 | 3624 | 46.8 (23.8–69.8) | 99.58 (<0.001) | Fig A1.12 in (S1 Fig) |
| | Vietnam | 1 | 35 | 173 | 20.2 (14.2–26.2) | NA | NA |
| Anxiety | All | 117 | 38284 | 99639 | 39.7(34.3–45.1) | 99.78(<0.001) | Fig 3 |
| | Asian Pacific region | 1 | 60 | 1146 | 5.2(3.9–6.5) | NA | NA |
| | Bangladesh | 2 | 414 | 782 | 52.1(21.5–82.7) | 98.81 (<0.001) | Fig A2.1 in (S2 Fig) |
| | China | 56 | 16605 | 54004 | 31.9 (27.8–36.0) | 99.27 (<0.001) | Fig A2.2 in (S2 Fig) |
| | Egypt and Saudi | 1 | 251 | 426 | 58.9(54.2–63.6) | NA | NA |
| | India | 10 | 1807 | 4098 | 44.2(32.6–55.9) | 98.41 (<0.001) | Fig A2.3 in (S2 Fig) |
| | Indonesia | 4 | 1122 | 2054 | 56.5(39.0–74.0) | 98.56 (<0.001) | Fig A2.4 in (S2 Fig) |
| | Iran | 3 | 794 | 1330 | 59.1(40.8–77.3) | 98.07 (<0.001) | Fig A2.5 in (S2 Fig) |
| | Iraq | 2 | 470 | 889 | 51.8(30.2–73.4) | 97.78 (<0.001) | Fig A2.6 in (S2 Fig) |
| | Jordan | 1 | 823 | 1163 | 70.8(68.2–73.4) | NA | NA |
| | Korea | 1 | 23 | 115 | 20.0(12.7–27.3) | NA | NA |
| | Malaysia | 2 | 438 | 1449 | 30.2(27.9–32.6) | 0 (0.493) | Fig A2.7 in (S2 Fig) |
| | Nepal | 2 | 343 | 879 | 38.8(32.7–44.9) | 72.36 (0.06) | Fig A2.8 in (S2 Fig) |
| | Oman | 2 | 659 | 1541 | 50.7(18.0–83.4) | 99.33 (<0.001) | Fig A2.9 in (S2 Fig) |
| | Pakistan | 6 | 4732 | 11372 | 52.4(31.9–72.9) | 99.45 (<0.001) | Fig A2.10 in (S2 Fig) |
| | Philippines | 2 | 525 | 1061 | 46.3(29.9–62.8) | 96.23 (<0.001) | Fig A2.11 in (S2 Fig) |
| | Qatar | 1 | 53 | 127 | 41.7(33.2–50.3) | NA | NA |
| | Saudi Arabia | 8 | 4988 | 8426 | 45.7(31.9–59.5) | 99.25 (<0.001) | Fig A2.12 in (S2 Fig) |
| | Singapore | 2 | 1308 | 3197 | 40.9(39.2–42.6) | 0 (0.345) | Fig A2.13 in (S2 Fig) |
| | Singapore and India | 1 | 142 | 906 | 15.7(13.3–18.0) | NA | NA |
| | Turkey | 10 | 3065 | 5289 | 48.9(27.0–70.7) | 99.8 (<0.001 | Fig A2.14 in (S2 Fig) |
| | Vietnam | 1 | 58 | 173 | 33.5(26.5–40.6) | NA | NA |

*(Continued)*

**Table 2.** (Continued)

| Domain | Country | N | Total Psychological distress | Total sample size | Prevalence,% (95% CI) | I² (p-value) | Appendix |
|---|---|---|---|---|---|---|---|
| Stress | All | 40 | 8599 | 34010 | 36.4 (23.2–49.7) | 99.45 (<0.001) | Fig 4 |
| | China | 8 | 2876 | 8552 | 29.3 (12.5–46.1) | 99.75 (<0.001) | Fig A3.1 in (S3 Fig) |
| | Egypt and Saudi | 1 | 238 | 426 | 55.9 (51.2–60.6) | NA | NA |
| | India | 7 | 1078 | 1845 | 49.4 (23.8–74.9) | 99.51 (<0.001) | Fig A3.2 in (S3 Fig) |
| | Indonesia | 2 | 570 | 1326 | 43.3 (20.5–66.1) | 98.71 (<0.001) | Fig A3.3 in (S3 Fig) |
| | Iran | 1 | 217 | 217 | 99.8 (99.1–100.4) | NA | NA |
| | Jordan | 1 | 287 | 448 | 64.1 (59.6–68.5) | NA | NA |
| | Korea | 1 | 5 | 115 | 4.3 (0.6–8.1) | NA | NA |
| | Malaysia | 2 | 363 | 1449 | 26.0 (20.6–31.4) | 77.65 (0.034) | Fig A3.4 in (S3 Fig) |
| | Nepal | 1 | 69 | 404 | 17.1 (13.4–20.7) | NA | NA |
| | Oman | 2 | 486 | 1541 | 38.6 (9.5–67.7) | 99.12 (<0.001) | Fig A3.5 in (S3 Fig) |
| | Pakistan | 4 | 635 | 10790 | 33.6 (-6.4–73.6) | 99.89 (<0.001) | Fig A3.6 in (S3 Fig) |
| | Qatar | 1 | 39 | 127 | 30.7 (22.7–38.7) | NA | NA |
| | Saudi Arabia | 3 | 128 | 498 | 29.9 (0.2–59.6) | 99.31 (<0.001) | Fig A3.7 in (S3 Fig) |
| | Singapore | 1 | 205 | 3075 | 6.7 (5.8–7.5) | NA | NA |
| | Singapore and India | 1 | 47 | 906 | 5.2 (3.7–6.6) | NA | NA |
| | Turkey | 3 | 1334 | 2118 | 46.9 (6.2–87.5) | 99.76 (<0.001) | Fig A3.8 in (S3 Fig) |
| | Vietnam | 1 | 22 | 173 | 12.7 (7.8–17.7) | NA | NA |
| Burnout | All | 7 | 5426 | 8732 | 68.3 (54.0–82.5) | 99.5 (<0.001) | Fig 5 |
| | Asian Pacific region | 1 | 182 | 301 | 60.5(54.9–66.0) | NA | NA |
| | China | 3 | 2859 | 5025 | 58.0 (30.5–85.6) | 99.78 (<0.001) | Fig A4.1 in (S4 Fig) |
| | Korea | 1 | 104 | 115 | 90.4(85.1–95.8) | NA | NA |
| | Malaysia | 1 | 184 | 216 | 85.2(80.4–89.9) | NA | NA |
| | Singapore | 1 | 2097 | 3075 | 68.2(66.5–69.8) | NA | NA |
| Fear | All | 5 | 3460 | 4302 | 71.3 (54.6–88.0) | 99.83 (<0.001) | Fig 6 |
| | Bangladesh | 1 | 370 | 370 | 99.9 (99.5–100.2) | NA | NA |
| | China | 2 | 1890 | 2353 | 53.4 (-20.6–127.5) | 99.93 (<0.001) | Fig A5.1 in (S5 Fig) |
| | Singapore | 1 | 89 | 122 | 73.0 (65.1–80.8) | NA | NA |
| | Turkey | 1 | 1111 | 1457 | 76.3 (74.1–78.4) | NA | NA |
| Low resilience | All | 2 | 78 | 484 | 16.1(12.8–19.4) | 0(0.922) | Fig 7 |
| | China | 1 | 59 | 364 | 16.2(12.4–20.0) | NA | NA |
| | India | 1 | 19 | 120 | 15.8(9.3–22.4) | NA | NA |

The pooled prevalence of mild to severe anxiety among HCPs in Asian countries was 39.7% (95% CI = 34.3–45.1). (Fig 3) Jordan, again recorded the highest prevalence of anxiety at 70.8% (95% CI = 68.2–73.4) among HCPs while the lowest prevalence of anxiety was found in a study conducted across the Asian Pacific region. Prevalence of anxiety in China was 31.9% (95% CI = 27.8–36.0), which is almost similar with Malaysia at 30.2% (95% CI = 27.9–32.6).

The overall pooled prevalence of mild to severe stress was found to be 36.4% (95% CI = 23.2–49.7) (Fig 4), with the highest prevalence reported in Iran at 99.8% (95% CI = 99.1–100.4) whereas the lowest prevalence was reported in Korea at 4.3% (95% CI = 0.6–8.1).

Pooled prevalence of moderate to severe burnout among HCPs in Asia was 68.3% (95% CI = 54.0–82.5) (Fig 5) which was relatively higher as compared to others mental health issues aforementioned. Korea was ranked number one for the pooled prevalence of burnout at 90.4% (95% CI = 85.1–95.8), followed by Malaysia at 85.2% (95% CI = 80.4–89.9) and Singapore at

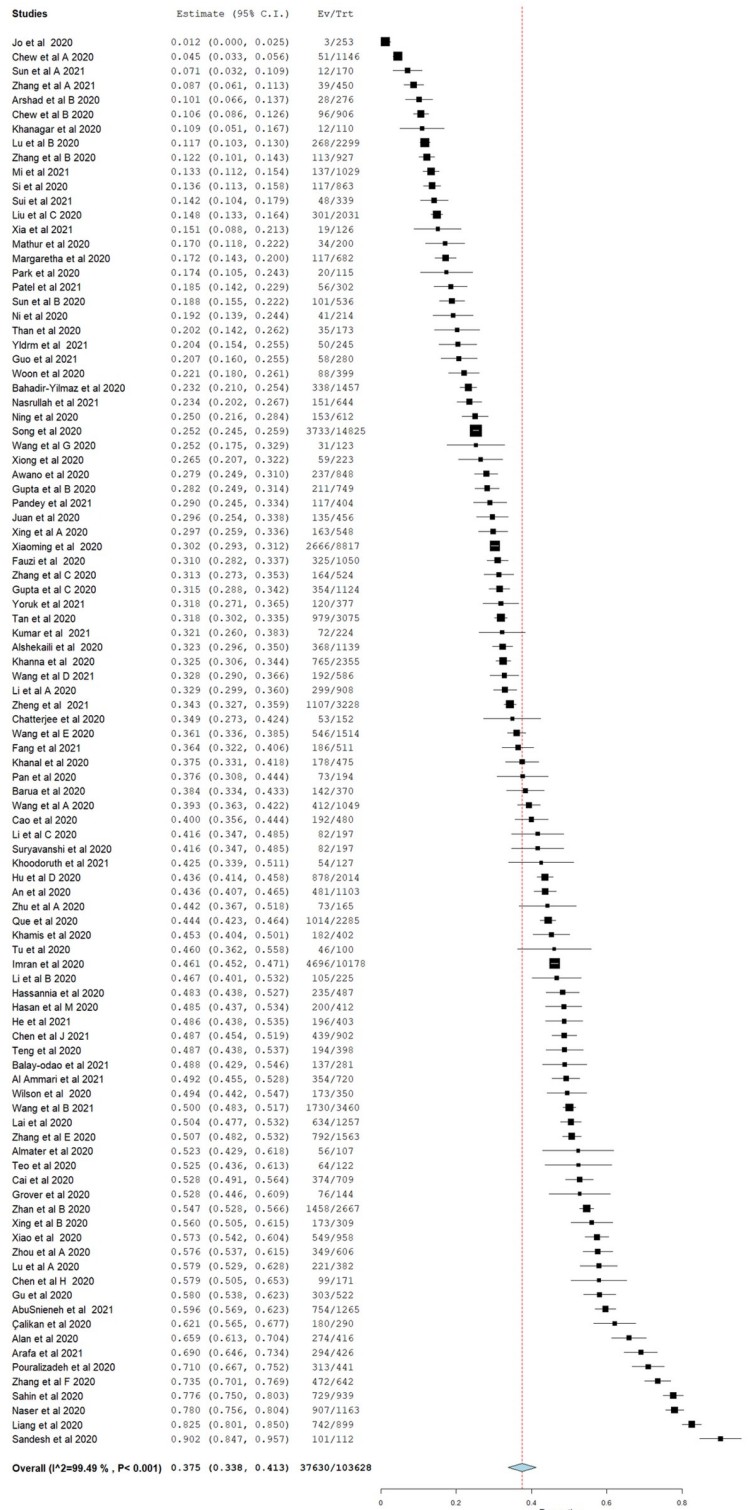

**Fig 2. Forest plot of overall pooled prevalence of depression.**

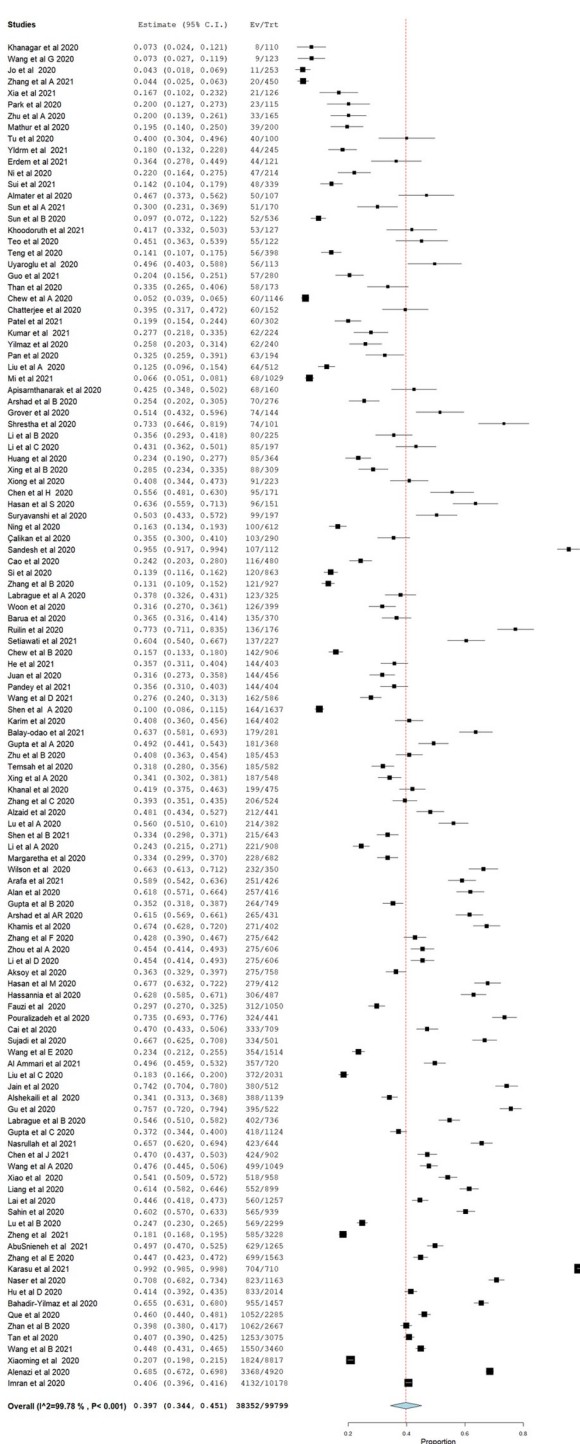

**Fig 3. Forest plot of overall pooled prevalence of anxiety.**

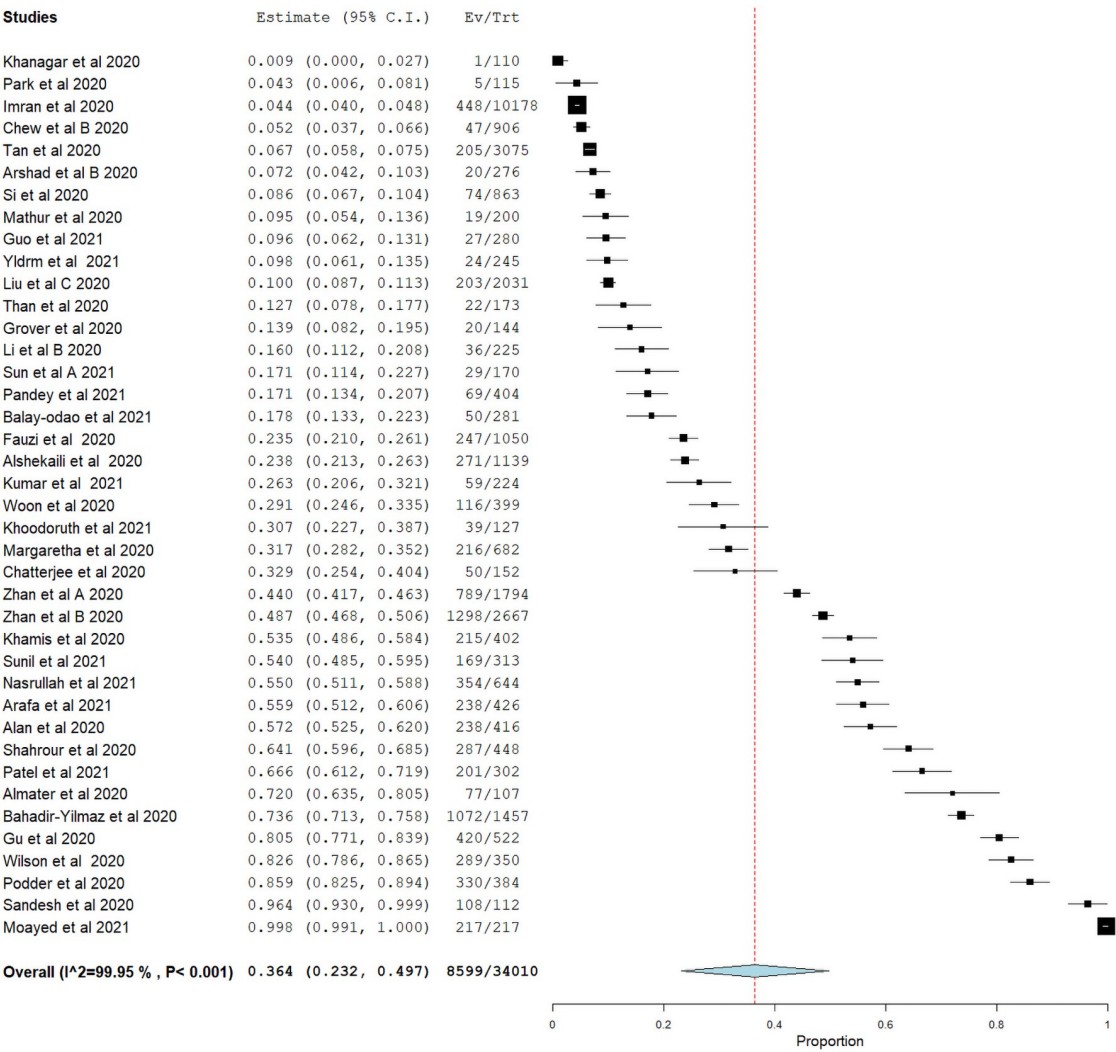

**Fig 4. Forest plot of overall pooled prevalence of stress.**

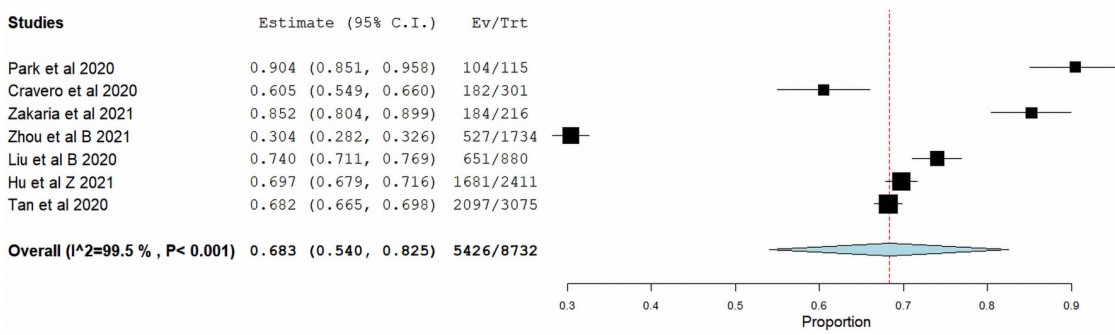

**Fig 5. Forest plot of overall pooled prevalence of burnout.**

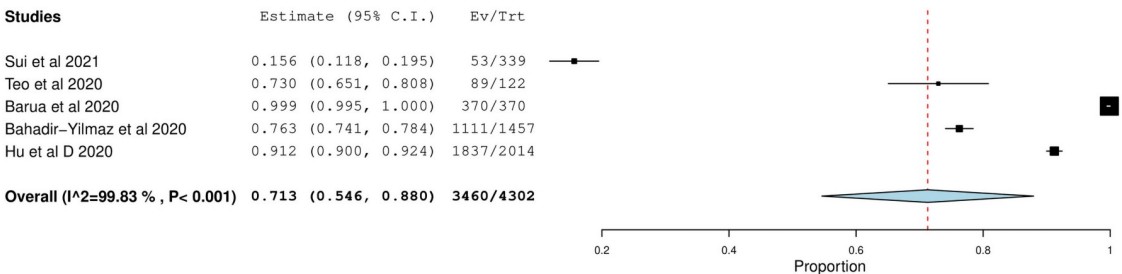

**Fig 6. Forest plot of overall pooled prevalence of fear.**

68.2% (95% CI = 66.5–69.8); while China had the lowest prevalence at 58.0% (95% CI = 30.5–85.6) among the 5 countries that were investigated for the prevalence of burnout.

The pooled prevalence of mild to severe fear was 71.3% (95% CI = 54.6–88.0) (Fig 6). The highest prevalence of fear was found in Bangladesh at 99.9% (95% CI = 99.5–100.2) while the lowest prevalence was found in China at 53.4% (95% CI = -20.6–127.5).

Low resilience had the pooled prevalence of 16.1% (95% CI = -12.8–19.4) (Fig 7), with 16.2% (95% CI = 12.4–20.0) and 15.8% (95% CI = 9.3–22.4) being reported in China and India, respectively.

## Subgroup analysis for the prevalence of psychological distress among healthcare providers amidst COVID-19 pandemic according to gender and occupation

Table 3 summarized the subgroup analysis of pooled prevalence of mental illness among healthcare providers during COVID-19 pandemic according to gender and occupation. The pooled prevalence of depression was higher in females (40.9%; 95%CI = 33.4–48.4) than males (35.5%; 95%CI = 29.5–41.6). Among all the HCPs, the nurses (39.3%; 95%CI = 33.2–45.3) had the highest prevalence of depression when compared to doctors (36.4%; 95%CI = 30.6–42.3) and allied healthcare personals (34.3%; 95%CI = 23.5–45.1) (S1 Fig).

Similarly females (50.6%; 95%CI = 43.5–57.6) generally were more anxious than males (41.2%; 95%CI = 32.0–50.4) while analysing the gender subgroup for anxiety. Nurses, assistant nurses and midwives (43.1%; 95%CI = 36.6–49.7) had the highest prevalence for anxiety, followed by doctors and dentists (39.6%; 95%CI = 34.5–44.7), and finally by allied healthcare personals and pharmacists (38.6%; 95%CI = 26.2–51.0).

For the prevalence on stress, the female population (48.1%; 95%CI = 31.6–64.5) was still having higher prevalence than male population (40.4%; 95%CI = 22.8–57.9). Almost half of those who worked as nurses, assistant nurses and midwifes (45.4%; 95%CI = 29.4–61.4)

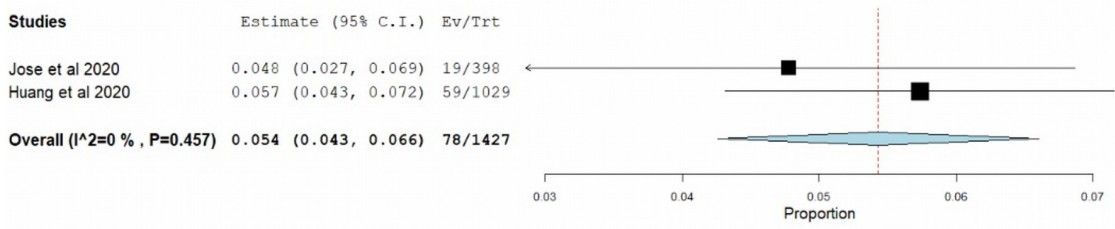

**Fig 7. Forest plot of the overall pooled prevalence of low resilience.**

**Table 3. Subgroup analysis of pooled prevalence of psychological distress among healthcare providers during COVID-19 pandemic according to gender and occupation.**

| Domain | Subgroup | N | Total Psychological distress | Total sample size | Prevalence,% (95% CI) | $I^2$ (p-value) | Appendix |
|---|---|---|---|---|---|---|---|
| Depression | Female | 29 | 7432 | 17595 | 40.9 (33.4–48.4) | 99.12 (<0.001) | Fig A1.13 in (S1 Fig) |
| | Male | 28 | 3883 | 10556 | 35.5 (29.5–41.6) | 97.49 (<0.001) | Fig A1.14 in (S1 Fig) |
| | Doctor/Dentist | 35 | 9679 | 23488 | 36.4 (30.6–42.3) | 98.79 (<0.001) | Fig A1.15 in (S1 Fig) |
| | Nurse/Assistant nurse/midwife | 38 | 8913 | 21725 | 39.3 (33.2–45.3) | 98.98 (<0.001) | Fig A1.16 in (S1 Fig) |
| | Allied healthcare personal/ Pharmacist | 17 | 966 | 2578 | 34.3 (23.5–45.1) | 97.96 (<0.001) | Fig A1.17 in (S1 Fig) |
| Anxiety | Female | 34 | 9359 | 19315 | 50.6 (43.5–57.6) | 99.07 (<0.001) | Fig A2.15 in (S2 Fig) |
| | Male | 33 | 5288 | 12434 | 41.2 (32.0–50.4) | 99.21 (<0.001) | Fig A2.16 in (S2 Fig) |
| | Doctor/Dentist | 41 | 9942 | 23846 | 39.6 (34.5–44.7) | 98.37 (<0.001) | Fig A2.17 in (S2 Fig) |
| | Nurse/Assistant nurse/midwife | 42 | 10445 | 24679 | 43.1 (36.6–49.7) | 99.25 (<0.001) | Fig A2.18 in (S2 Fig) |
| | Allied healthcare personal / Pharmacist | 19 | 2279 | 4302 | 38.6 (26.2–51.0) | 98.70 (<0.001) | Fig A2.19 in (S2 Fig) |
| Stress | Female | 14 | 1752 | 9812 | 48.1 (31.6–64.5) | 99.78 (<0.001) | Fig A3.9 in (S3 Fig) |
| | Male | 14 | 957 | 6393 | 40.4 (22.8–57.9) | 99.62 (<0.001) | Fig A3.10 in (S3 Fig) |
| | Doctor/Dentist | 15 | 1698 | 13938 | 33.5 (22.7–44.2) | 99.60 (<0.001) | Fig A3.11 in (S3 Fig) |
| | Nurse/Assistant nurse/midwife | 12 | 4229 | 9142 | 45.4 (29.4–61.4) | 99.68 (<0.001) | Fig A3.12 in (S3 Fig) |
| | Allied healthcare personal / Pharmacist | 5 | 149 | 568 | 31.4 (15.7–47.1) | 95.35 (<0.001) | Fig A3.13 in (S3 Fig) |
| Burnout | Doctor/Dentist | 5 | 1115 | 1575 | 74.9 (62.8–87.1) | 95.19 (<0.001) | Fig A4.2 in (S4 Fig) |
| | Nurse/Assistant nurse/midwife | 2 | 1012 | 1431 | 80.2 (56.8–103.7) | 98.83 (<0.001) | Fig A4.3 in (S4 Fig) |
| | Allied healthcare personal / Pharmacist | 1 | 24 | 37 | 64.9 (49.5–80.2) | NA | NA |
| Fear | Male | 1 | 233 | 233 | 99.8 (99.2–100.4) | NA | NA |
| | Female | 1 | 147 | 147 | 99.7 (98.7–100.6) | NA | NA |
| | Doctor/Dentist | 1 | 370 | 370 | 99.9 (99.5–100.2) | NA | NA |
| | Nurse/Assistant nurse/midwife | 3 | 3001 | 3810 | 61.1 (28.0–94.1) | 99.85 (<0.001) | Fig A5.2 in (S5 Fig) |
| | Allied healthcare personal / Pharmacist | 1 | 89 | 122 | 73.0 (65.1–80.8) | NA | NA |
| Low resilience | Female | 1 | 15 | 88 | 17.0 (9.2–24.9) | NA | NA |
| | Male | 1 | 4 | 32 | 12.5 (1.0–24.0) | NA | NA |
| | Nurse/Assistant nurse/midwife | 1 | 19 | 120 | 15.8 (9.3–22.4) | NA | NA |

experience stress, while about one third of doctors or dentists (33.5%; 95%CI = 22.7–44.2), and allied healthcare personals or pharmacists (31.4%; 95%CI = 15.7–47.1), respectively also experienced stress.

For the data on burnout, nurses population (80.2%; 95%CI = 56.8–103.7) remained at the top position in terms of experiencing burnout, followed by the doctors population (74.9%; 95%CI = 62.8–87.1) and finally by allied healthcare personals (64.9%; 95%CI = 49.5–80.2).

For the prevalence of fear, both genders were deemed to have almost similar prevalence with females (99.7%; 95%CI = 98.7–100.6) and males (99.8%; 95%CI = 99.2–100.4). Doctors (99.9%; 95%CI = 99.5–100.2) were having the highest prevalence of fear while nurses had the lowest prevalence (61.1%; 95%CI = 28.0–94.1). Besides, females (17.0%; 95%CI = 9.2–24.9) reported a higher prevalence of having low resilience as compared to males (12.5%; 95% CI = 1.0–24.0); with the prevalence of low resilience among nurses was 15.8% (95%CI = 9.3–22.4).

## Subgroup analysis for risk factors associated with psychological distress among healthcare providers amidst COVID-19 pandemic

Table 4 showed the subgroup analysis of the odds ratio of psychological distress according to the risk factors. From this meta-analysis, it was found that odds of depression were increased in females (OR = 1.48; 95%CI = 1.30–1.68) and those who worked as nurses, assistant nurses or midwives (OR = 1.21; 95% CI = 1.02–1.45). Those who worked as doctors, dentist, allied healthcare or pharmacist were found to be the protected against depression (p-value <0.001).

In terms of the risk of getting anxiety, females (OR = 1.66; 95%CI = 1.49–1.85) and nurses, assistant nurse or midwife (OR = 1.36; 95%CI = 1.16–1.58) had a higher risk than doctors, dentist (OR = 0.82; 95%CI = 0.73–0.93), allied healthcare personals and pharmacists (OR = 0.89; 95%CI = 0.74–1.06), with a p-value of <0.001. Besides, females were also a risk factor for the development of stress (OR = 1.59; 95%CI = 1.28–1.97).

## Sensitivity analysis and publication bias

We performed sensitivity analysis by omitting every single study step-by step from the meta-analytic model. The result reported no major changes in terms of the pooled prevalence of psychological distress (Fig A6.1-A6.6 in S6 Fig). The visual assessment of the funnel plot for all the psychological distress parameters showed a high publication bias (Fig A7.1-A7.5 in S7 Fig), which was confirmed by Egger's test for depression and anxiety.

## Discussion

To date, this is the first systematic review with meta-analysis on psychological distress among HCPs amidst the COVID-19 pandemic in Asian region with country-based estimates. We identified 148 cross-sectional studies from 23 Asia countries and quantitatively determined the subgroup pooled prevalence according to gender and occupations in this region. The pooled prevalence for depression was 37.5% (95%CI: 33.8–41.3), anxiety 39.7(95%CI: 34.3–45.1), stress 36.4% (95%CI: 23.2–49.7), fear 71.3% (95%CI: 54.6–88.0), burnout 68.3% (95%CI: 54.0–82.5), and low resilience was 16.1% (95%CI: 12.8–19.4), respectively.

Among all the psychological distress, fear appeared to be the most common psychological reaction among HCPs whom continued to provide healthcare services during the COVID-19 pandemic, followed by burnout, anxiety, depression and finally by stress. More than two thirds of the HCPs were having fear amidst the COVID-19 pandemic. Our findings is similar to the findings of the previous review in Asia which the prevalence of fear among HCPs ranged between 67% [166] to 77.1% [13]. There are many reason for HCW's fear in this pandemic,

**Table 4. Subgroup analysis of odd ratio of psychological distress among HCPs according to gender and occupations.**

| Subgroup analysis | N | Exposure in at risk group | Total at risk group | Exposure in control | Total control | OR (95%CI) | $I^2$ (p-value) | Appendix |
|---|---|---|---|---|---|---|---|---|
| **Depression** | | | | | | | | |
| Gender: Female | 29 | 7432 | 17595 | 3789 | 10556 | 1.48 (1.30–1.68) | 68.10 (<0.001) | Fig A1.18 in (S1 Fig) |
| Occupation Doctor/ Dentist | 22 | 2889 | 7329 | 3334 | 9109 | 0.87(0.69–1.10) | 86.47 (<0.001) | Fig A1.19 in (S1 Fig) |
| Nurse/Assistant nurse/midwife | 24 | 2716 | 7142 | 3791 | 9812 | 1.21 (1.02–1.45) | 79.11 (<0.001) | Fig A1.20 in (S1 Fig) |
| Allied healthcare personal/ pharmacist | 16 | 902 | 2456 | 3612 | 8764 | 0.93 (0.69–1.25) | 84.75 (<0.001) | Fig A1.21 in (S1 Fig) |
| **Anxiety** | | | | | | | | |
| Female | 33 | 9319 | 19215 | 5288 | 12434 | 1.66(1.49–1.85) | 61.68(<0.001) | Fig A2.20 in (S2 Fig) |
| Doctor/ Dentist | 23 | 3546 | 8474 | 6135 | 12404 | 0.82(0.73–0.93) | 62.85(<0.001) | Fig A2.21 in (S2 Fig) |
| Nurse/Assistant nurse/midwife | 24 | 4002 | 8561 | 5690 | 12456 | 1.36(1.16–1.58) | 75.58(<0.001) | Fig A2.22 in (S2 Fig) |
| Allied healthcare personal /pharmacist | 16 | 2144 | 3955 | 5642 | 11328 | 0.89(0.74–1.06) | 68.78(<0.001) | Fig A2.23 in (S2 Fig) |
| **Stress** | | | | | | | | |
| Female | 14 | 1752 | 9812 | 957 | 6393 | 1.59 (1.28–1.97) | 56.14 (0.005) | Fig A3.14 in (S3 Fig) |
| Doctor/ Dentist | 6 | 481 | 1571 | 817 | 2668 | 0.80 (0.43–1.49) | 87.67 (<0.001) | Fig A3.15 in (S3 Fig) |
| Nurse/Assistant nurse/midwife | 6 | 704 | 2325 | 594 | 1914 | 1.47 (0.80–2.70) | 87.02 (<0.001) | Fig A3.16 in (S3 Fig) |
| Allied healthcare personal /pharmacist | 4 | 113 | 343 | 562 | 1343 | 0.85 (0.61–1.17) | 20.27 (0.288) | Fig A3.17 in (S3 Fig) |
| **Burnout** | | | | | | | | |
| Female | 0 | | | | | | | |
| Doctor/ Dentist | 1 | 800 | 1122 | 881 | 1289 | 1.15(0.97–1.37) | Not sig | |
| Nurse/Assistant nurse/midwife | 2 | 1012 | 1431 | 853 | 1196 | 1.93(0.37–10.12) | 93.96(<0.001) | Fig A4.4 in (S4 Fig) |
| Allied healthcare personal /pharmacist | 1 | 24 | 37 | 160 | 179 | 0.22(0.10–0.50) | Significant-NA | |
| **Fear** | | | | | | | | |
| Female | 1 | 147 | 147 | 223 | 223 | 0.66 (0.01–33.44) | Not sig | |
| Doctor/ Dentist | 0 | | | | | | | |
| Nurse/Assistant nurse/midwife | 0 | | | | | | | |
| Allied healthcare personal /pharmacist | 0 | | | | | | | |
| **Low resilience** | | | | | | | | |
| Female | 1 | 15 | 88 | 4 | 32 | 1.44(0.44–4.71) | NA | |
| Doctor/ Dentist | 0 | | | | | | | |
| Nurse/Assistant nurse/midwife | 0 | | | | | | | |
| Allied healthcare personal /pharmacist | 0 | | | | | | | |

one of the most common issues face by HCPs are there are fear of failing to provide adequate care for patients, non-limited to only COVID-19 due to limited supply of resources as well as manpower to handle the frequent sudden rise in COVID-19 cases that often strain the health-care facility capacity [167]. Another possible explanation for the high prevalence of fear among HCPs were due to the fear of carrying the virus back home and infecting family and friends as well as fear of stigmatization [168]. It is also could be due to the fear of being infected and need to be quarantined, thereby further exacerbating the pre-existing inadequate numbers of HCWs at the frontline combating COVID-19 [169]. Another reason for fear among HCPs was because they were lacking in Personal Protection Equipment (PPV) rations and had unfamiliarity in using PPV, especially at the beginning of the pandemic. They may have had no training for infection prevention and control protocols especially at the beginning of the pandemic. All these factors may end up with many HCPs succumbing to this virus while providing care to the patients [170, 171].

Burnout was the second most common psychological distress faced by HCPs. The possible explanation could be due to the irregular or long working hours and high workload demand [1]. Furthermore, HCWs need to adapt to the IPC strategies by putting on PPE before starting to work and showering before going home. All this required additional time in preparation and cleaning and can cause fatigue in the long run [172]. This is not surprising on why another systematic review by de Pablo et al., 2020 reported that only 34.4%(95%CI = 19.3–53.5%) of the HCPs suffered from burnout which is lower compared with our finding of 68.3% [172]. The possible explanation for the difference in prevalence of burnout could be due to the fact that the study by de Pablo et al., 2020 examined burnout among HCWs exposed to SARS/MERS/COVID-19 whereas our systematic focused mainly on COVID-19. Despite SARS/MERS/COVID-19 are all caused by different strains of coronavirus, however there is an apparent difference in the influence on human imposed by these different coronaviruses' strains. For instance, the mortality rate of COVID-19 is 4.9%, which is higher than SARS (0.96%) and much lower than MERS (34.4%). The duration of the SARS pandemic (cumulative of 8,422 cases) was relative shorter which was brought under control in only 9 months (1 November 2002–31 July 2003) and outbreak of MERS only lasted for two months in both Saudi Arabia (n = 402 cases) and South Korea (n = 150 cases). However, COVID-19 has been around us for more than one year now and the emergence of new variants that are still evolving and its effect largely remains unpredictable, adds to its own pandora box. In view of the influence exerted by COVID-19 is much more devastating than SARS and MERS, therefore HCWs suffer burnout more easily in handing COVID-19 cases than in handling cases of SARS and MERS.

Our review showed that about one third of the HCPs suffered from depression with pooled prevalence of 37.5%, anxiety (39.7%) and stress (36.4%). However, our pooled prevalence of depression was higher comparing with two other systematic reviews by Hossain MM et al and Gonzalo Salazar de Pablo et al where the pooled prevalence of depression ranges from 17.9%-29.9% and anxiety, 22.2% to 43.6% [12, 171]. The pooled prevalence of anxiety in our review fell in between of these two reviews which ranged from 22.2 to 43.6% [12, 171].

Among the 148 papers reviewed, 98 of them examined the prevalence of depression among HCP, with China reporting most number of cases (22,772) of depression since the outbreak of COVID-19. Prevalence of this parameter in China was found to be 36.5%. Prevalence of depression in Malaysian HCP was relatively lower at 26.6% (95%CI: 17.9–35.3; p-value<0.001). The possible reason could be due to the fact that these cross-sectional studies were conducted at the very beginning stage of the outbreak in China before the pandemic was declared, where very little was known about the virus and hence the Chinese healthcare workers were generally experiencing greater mental disturbances.

Jordan, Egypt from the subdivision of Asian continent, Arab Saudi and Iran were the countries with the prevalence of depression and, anxiety of more than 50% when compared with other countries in Asia. The possible explanation could be due to the fact that most of the research were conducted between March to July 2021 at the time that Middle East countries were having the COVID-19crisis. Most of the countries (Jordan, Egypt, Arab Saudi and Iran) had full lockdown or night-time curfew. The reason for the rapid rise in cases were largely due to large religious gatherings, wedding celebrations and other social events where control measures were not sufficiently enforced [173]. The possible reasons were that there was a lack ofhealthcare facilities and equipment to deal with COVID-19 pandemic [174]. Furthermore, the human resources were also insufficient and below the recommendation of WHO as some of HCPs have left the countries following the country's own politic instability [175, 176]. With a weakened healthcare system, the COVID-19outbreak had posed a major challenge on the mental health of HCPs and it explained why the prevalence of depression and anxiety were generally higher as compared with other countries in Asia [177].

On the other hand, we also found that the prevalence of anxiety among healthcare workers in China, India, and Malaysia were 31.9% (95%CI: 27.8–36.0), 44.2% (95%CI: 32.6–55.9) and 30.2% (95%CI: 27.9–32.6), respectively. The lower prevalence of both anxiety and depression in Malaysia as compared with other countries could be due to the studies were conducted during Conditional Movement Control Order (CMCO) period when the condition of outbreak in Malaysia was considered to be relatively under-control, and background of the causative virus had been learnt from China. Besides, the fact that the data was not merely focusing on COVID-19 hospitals but also involving non-COVID centres could have led to the lower anxiety prevalence in Malaysia [140, 142].

With the COVID-19 pandemic hitting many health care facilities that were unprepared to handle it, many healthcare providers who were standing at the frontline were working and pushing themselves to the limit [28]. Pooled prevalence of burnout was found to be 68.3% (95% CI: 54.0–82.5) in our systematic review and meta-analysis, ranging from the lowest prevalence of 58.0% in China to the highest prevalence of 90.4% reported in Korea. Korea has the highest prevalence of burnout as the study was conducted specifically among the the Infectious Disease physicians. This is not unexpected as their work burden is much higher during the COVID-19 pandemic [160].

A cross-sectional study conducted by Dong et al in China revealed that despite the long working hours, healthcare workers were mostly (n = 4,120, 89.2%) motivated and feeling positive towards their task at hand, and remained committed to their professions. During the outbreak period in China around early 2020, Chinese nationals from all over the country, with or without medical background, showed exemplary courage and actively volunteered to assist at Wuhan, the epitome of the COVID-19 pandemic [178, 179]. There was also an increment in healthcare providers' salaries by the Chinese government at the same time. Their firm belief to their professions and strong social support from their nation were believed to be factors contributing to the lowest burnout prevalence among Asian countries [28]. Whereas, in Malaysia, the high prevalence of burnout among nurses could be due to the reduction in their off days as a consequences of more intense shift hours, and being overworked, coupled with a low salary [142].

When analysing the subgroup of associated risk factors of psychological distress, it was found that both females and nurses population were more at risk of getting mental distress such as depression [(Female: OR = 1.48; 95% CI = 1.30–1.68), (Nurse: OR = 1.21; 95% CI = 1.02–1.45)], anxiety [(Female: OR = 1.66; 95% CI = 1.49–1.85), (Nurse: OR = 1.36; 95% CI = 1.16–1.58)], and stress [(Female: OR = 1.59; 95%CI = 1.28–1.97), (Nurse: OR = 1.47; 95% CI = 0.80–2.70)]. Internally, females' nature generally belongs to the sentimental type and they

usually experience hormonal changes which would then affect their mood and emotion [180]. Other than playing a role as a medical professional, most of the time females were also house-wives for their family [8]. They tend to be a multitasker where they must take care of their family members' health and well-being as well as going out for marketing in crowded areas to purchase necessities. A lot of time they were lacking support from family and were bombarded with many negative news circulated on social media, which can create negative feelings and make them more tired and prone to psychological illness [181–183]. Besides, it was found that the majority of the nurses were females (95.6%) [184]. Furthermore, nurses generally had longer contact hours with COVID-19 patients than doctors and were working longer hours than usual [36]. Other than taking care of the patients, they had to deal with their families who might be more frustrated, angry, anxious or worried due to lack in family time [142]. These were consistent with the findings from a systematic review by Thatrimontrichai et al which concluded that females, nurses, having direct contact with infected patients, working longer hours and possessing less working experience were the main risk factors leading to mental disturbances among Chinese healthcare workers [13].

High heterogeneity was found in this systematic review, and possible reasons include variation from the participants' characteristics, outcome level and research setting [185]. Firstly, the high heterogeneity for the overall prevalence of psychological distress found in this study could be due to differences in the screening methods and diagnostic criteria in different countries with different ethnicities and research settings (S4 Table). For example, different screening tools were used to determine burnout such as Maslach Burnout Inventory (MBI) ProQOL Scale of Chinese Version and Olenburg Burnout Inventory (OLBI); whereas scales like Patient Health Questionnaaire (PHQ), Depression-Anxiety-Stress Scale-21 (DASS-21), Beck Depression Inventory (BDI), Hamilton depression rating scale (HAMD), etc were used for depression screening. Next, different categories for the severity of disease were reported in different studies. Moreover, even though in the same country, there were different diagnostic criteria applied for psychological distress studies. For example, in China, ten different diagnostic tools were used to detect depression compared with three diagnostic tools used in India, which give rise to a broad range of prevalence of depression in China (31.7%-41.2%) and India (27.9%-39.3%). In our meta-analysis, there was a wide gap in terms of sample size in all 148 studies, ranged from 100 to 14,825 in this review. All these factors would explain why there was a high heterogeneity for the prevalence of psychological distress, for instance, depression in this study.

Multiple factors contributed to publication bias, comprises of study design, sample size, decision of authors, journal editors and reviewers [186]. We had excluded all the unpublished data and studies with a sample size of below 100 in this review. However, we need to weigh the advantage and disadvantages as those published studies have gone through a rigorous review process, which gave a more reliable result compared with those unpublished data. Furthermore, we also excluded studies with outcomes that did not fulfil our operational definition of psychological distress. This systematic only included manuscripts wrote in English due to constraint of resources.. Thus, we had to interpret the results of this systematic review carefully within the context of its limitations level and research setting [185].

## Strengths and limitations

This review paper highlights the psychological distress of HCPs in the menacing era of the COVID-19 pandemic. Such HCW's mental burden either have not been acknowledged or have been underestimated because most of the healthcare systems have currently focused on coping with the pandemic as its main target. Foremost,psychological distress among HCPs

should be given priority and urgent action is needed to reduce the psychological impacts on HCPs, in order to ensure continuation of effective services to patients amidst the COVID-19 pandemic.

There are some limitations in this review paper. Firstly,our review paper depended on collecting and compiling the published data where the papers that were analysed were mostly periodic, in which only the psychological state of HCPs over a certain period of time were reflected. However, with the progression of time, different conditions of outbreak over the period of one year and shifting to new environments, the targeted population's mental health may have changed. Thus, the psychological impact among healthcare providers amidst the COVID-19 pandemic should ideally be assessed longitudinally. Secondly, high heterogeneity was not an unexpected finding in our review as data were gathered from various studies that were conducted differently in terms of study designs, data collection tools, different study setting and location as well as having varied demographic features of participants. Therefore, our results needs careful interpretation. Thirdly, only English language written articles were recruited in this review, therefore this can result in publication bias. Future studies on interventions to improve the psychological health of HCPs is needed urgently in order to maintain their physical health and productivity in continuing the fight against this pandemic.

## Conclusions

In conclusion, the global COVID-19 pandemic has had a devastating impact on the mental health of HCPs. This systematic review synthesizes the quantitative evidence of psychological distress among HCPs in Asian countries, which showed that one third of HCPs suffered from depression, anxiety and stress and more than two third of HCPs suffered from fear and burnout during the COVID-19 pandemic in Asia. Meta-analysis reported both females and nursse were at increased risk of having depression and anxiety. Female HCPs was also at a higher risk of getting stress when compared with the male HCPs. Urgent action are needed to implement a multicultural level interventions to support HCPs in order to reduce the burden of psychological distress during this very challenging COVID-19 pandemic.

## Supporting information

**S1 Fig. Forest plots of depression.**
(RAR)

**S2 Fig. Forest plots of anxiety.**
(RAR)

**S3 Fig. Forest plots of stress.**
(RAR)

**S4 Fig. Forest plots of burnout.**
(RAR)

**S5 Fig. Forest plots of fear.**
(RAR)

**S6 Fig. Leave-out-one Forest plots.**
(RAR)

**S7 Fig. Funnel plots.**
(RAR)

**S1 Table. PRISMA checklist.**
(DOCX)

**S2 Table. Search terms used from 13th of March to 15th of March 2021.**
(DOCX)

**S3 Table. Strobe checklist.**
(DOCX)

**S4 Table. Characteristics of 148 studies.**
(DOCX)

## Author Contributions

**Conceptualization:** Siew Mooi Ching, Ai Theng Cheong.

**Data curation:** Siew Mooi Ching, Kar Yean Ng, Hisham Ranita.

**Formal analysis:** Kai Wei Lee, Poh Ying Lim, Navin Kumar Devaraj.

**Methodology:** Kai Wei Lee, Poh Ying Lim.

**Resources:** Hisham Ranita.

**Software:** Pei Boon Ooi.

**Supervision:** Anne Yee, Ai Theng Cheong.

**Writing – original draft:** Siew Mooi Ching, Kar Yean Ng.

**Writing – review & editing:** Anne Yee, Navin Kumar Devaraj, Pei Boon Ooi, Ai Theng Cheong.

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
