## [Decision Letter · Decision Letter 0]

19 Aug 2021

PONE-D-21-21203

Psychological Distress Among Healthcare Providers During COVID-19 In Asia: Systematic Review and Meta-Analysis

PLOS ONE

Dear Dr. Cheong

Thank you for submitting your manuscript to PLOS ONE. After careful consideration, we feel that it has merit but does not fully meet PLOS ONE’s publication criteria as it currently stands. Therefore, we invite you to submit a revised version of the manuscript that addresses the points raised during the review process.

We look forward to receiving your revised manuscript.

Kind regards,

Ali Rostami

Academic Editor

PLOS ONE

2. Please remove your figures from within your manuscript file, leaving only the individual TIFF/EPS image files, uploaded separately.  These will be automatically included in the reviewers’ PDF.

3. We note that this manuscript is a systematic review or meta-analysis; our author guidelines therefore require that you use PRISMA guidance to help improve reporting quality of this type of study. Please upload copies of the completed PRISMA checklist as Supporting Information with a file name “PRISMA checklist

Reviewers' comments:

Reviewer's Responses to Questions

**Comments to the Author**

1. Is the manuscript technically sound, and do the data support the conclusions?

Reviewer #1: Yes

2. Has the statistical analysis been performed appropriately and rigorously? 

Reviewer #1: Yes

3. Have the authors made all data underlying the findings in their manuscript fully available?

Reviewer #1: Yes

4. Is the manuscript presented in an intelligible fashion and written in standard English?

Reviewer #1: Yes

5. Review Comments to the Author

Reviewer #1: The topic is interesting and contains useful information for readers, although the following comments will help reinforce this manuscript.

1- - In the abstract, name, the used databases.

2- - Please in introduction, highlighted which factors affect the psychological distress of HCW based on quantitative and especially qualitative studies conducted in vulnerable countries in Asia?

3- - exactly mention who do you mean by HCW?

4- Please do a search on the Web of Sciences database.

5- Please specify the inclusion and exclusion criteria clearly.

6- Did you do a manual search (based on Google Scholar citation or check the references of each article)

7- Please improve the quality of Forest plots (All Figures).

8- Please discuss the causes of high heterogeneity in this study (in discussion section).

9- Please explain the reason for publication bias in this study (in discussion section).

6. PLOS authors have the option to publish the peer review history of their article (what does this mean?). If published, this will include your full peer review and any attached files.

Reviewer #1: **Yes: **Maryam Azizi

---

## [Author Response · Author response to Decision Letter 0]

8 Sep 2021

Cheong Ai Theng

Associate Professor

Department of Family Medicine, Faculty of Medicine and Health

Science, Selangor, Malaysia

Date: 6 September 2021

Academic Editor:

Prof. Dr. Khatijah Lim Abdullah 

PLOS ONE

Dear Editor

RESUBMISSION OF MANUSCRIPT TO PLOS ONE (PONE-D-21-21203.R1)

I’m enclosing herewith a revised manuscript entitled “Psychological Distress Among Healthcare Providers During COVID-19 In Asia: Systematic Review and Meta-Analysis” for publication in PLOS ONE. 

All authors have read and approved the revised manuscript. Changes are made in blue-coloured text. Comments and suggestions from reviewers have been addressed clearly point by point following this covering letter. 

Thank you for your kind attention.

Sincerely,

Cheong Ai Theng…

Cheong Ai Theng, Ph D

Corresponding Author

Department of Family Medicine, 

Faculty of Medicine and Health Science, 43400 Serdang UPM, Selangor Malaysia

Tel: +6 (03) 9769 2538

Email: cheaitheng@upm.edu.my

 

Thank you for the pertinent comments and we have made the changes accordingly. 

Comment

1. In the abstract, name, the used databases.

Answer: Thank you for your comment, we have made the amendment to read as

“COVID-19 pandemic is having a devastating effect on the mental health and wellbeing of healthcare providers (HCPs) globally. This review aimed at determining the prevalence of depression, anxiety, stress, fear, burnout and resilience and its associated factors among HCPs in Asia during the COVID-19 pandemic. We performed literature search using 4 databases from Medline, Cinahl, PubMed and Scopus up to March 15, 2021 and selected cross-sectional studies.”

Comment 

2. Please in introduction, highlighted which factors affect the psychological distress of HCW based on quantitative and especially qualitative studies conducted in vulnerable countries in Asia?

Answer: We have highlighted the factors that affect the psychological distress of HCW in the Introduction section, under third paragraph to read as

“In performing their duties of arresting the spread of COVID-19, the HCP are risking their lives due to a higher risk of virus exposure, high workload demand, irregular or long working hours and increased psychological distress such as depression, anxiety, stress, occupation burnout, fear, low resilience as well as fatigue[1]. In addition, the HCP were barred from taking holiday and separated from their loved ones up to weeks or even months. Wearing the full personal protective equipment or gear (PPE) that is very uncomfortable for long hours continuously every day while managing patients diagnosed with COVID-19 is extremely exhausting, not to mention that this has become a routine task in their daily work. Literature reported that factors associated with personal-, work-, and patient-related burnout among HCPs were those had direct involvement in COVID-19 management, underlying medical illness, and receiving inadequate psychological support in the workplace [6]. Those with higher points in coping score were significantly associated with reduction in anxiety and depression score [7]. Other significant factors associated with psychological distress inluding but not limited to thought of resignation and reluctant to work, fear of infecting family member, frequent change in infection prevention and control protocol or guideline, and poor social support [8]. All of the aforementioned factors had been determined as factors that are leaving negative impacts on the healthcare workers in Asian countries psychologically[8-10]. In performing their duties of arresting the spread of COVID-19, the HCP are risking their lives due to a higher risk of virus exposure, high workload demand, irregular or long working hours and increased psychological distress such as depression, anxiety, stress, occupation burnout, fear, low resilience as well as fatigue[1]. In addition, the HCP were barred from taking holiday and separated from their loved ones up to weeks or even months. Wearing the full personal protective equipment or gear (PPE) that is very uncomfortable for long hours continuously every day while managing patients diagnosed with COVID-19 is extremely exhausting, not to mention that this has become a routine task in their daily work. Literature reported that factors associated with personal-, work-, and patient-related burnout among HCPs were those had direct involvement in COVID-19 management, underlying medical illness, and receiving inadequate psychological support in the workplace [6]. Those with higher points in coping score were significantly associated with reduction in anxiety and depression score [7]. Other significant factors associated with psychological distress inluding but not limited to thought of resignation and reluctant to work, fear of infecting family member, frequent change in infection prevention and control protocol or guideline, and poor social support [8]. All of the aforementioned factors had been determined as factors that are leaving negative impacts on the healthcare workers in Asian countries psychologically[8-10].”

Comment

3. exactly mention who do you mean by HCW?

Answer: We have added this statement under PICO to read as 

The participants should be HCPs (doctors, dentists, nurses, nurse assistants, midwives, medical assistants, pharmacists, and other allied healthcare workers).

Comment

4. Please do a search on the Web of Sciences database.

Answer: We are sorry to inform that our institute does not have legitimate website in assessing Web of Science. Furthermore, four database were adequate to capture all the relevant studies. 

Comment

5. Please specify the inclusion and exclusion criteria clearly.

Answer: We have revised the inclusion and exclusion criteria in the Methodology section, 4th paragraph to read as

Selection criteria

The inclusion criteria for this systematic review were as follow: 

a) The study design was cross-sectional with a minimum sample size of 100

b) The study stated the prevalence of depression, anxiety, stress, burnout, fear and resilience among HCPs during COVID-19 pandemic

c) The study evaluated depression, anxiety, stress, burnout, fear and resilience based on validated instrument tools or scales

d) The study involved HCPs (doctors, dentists, nurses, nurse assistants, midwives, medical assistants, pharmacists, and other allied healthcare workers) from Asian countries

e) The studies must be published in English peer-reviewed journal. 

Studies with the following criteria were excluded: 

a) Perspective, opinion, review articles, case reports, short communications paper, no full text study and unpublished data

b) Data reported in continuous or qualitative format

c) Outcomes were not clearly defined by validated tools

d) Depression, anxiety, stress, burnout, fear and resilience were reported as independent data

e) Technical error was present in the reported data

f) After full-text articles have been assessed for eligibility, those outcomes were grouped into category of severities which were different from our operational definition 

Comment

6. Did you do a manual search (based on Google Scholar citation or check the references of each article)

Answer: We didn’t perform manual search as there may not have add on value in view of there were 148 studies with big sample size recruited in this systematic review.

 

Comment

7. Please improve the quality of Forest plots (All Figures).

Answer: We have improved the quality of Forest plots by changing PNG to tiff with 300 dpi.

Comment

8. Please discuss the causes of high heterogeneity in this study (in discussion section).

Answer: We have added the causes of high heterogeneity in the discussion session, second last paragraph to read as:

High heterogeneity was found in this systematic review, and possible reasons include variation from the participants’ characteristics, outcome level and research setting [186]. Firstly, the high heterogeneity for the overall prevalence of psychological distress found in this study could be due to differences in the screening methods and diagnostic criteria in different countries with different ethnicities and research settings (S4 Table). For example, different screening tools were used to determine burnout such as Maslach Burnout Inventory (MBI) ProQOL Scale of Chinese Version and Olenburg Burnout Inventory (OLBI); whereas scales like Patient Health Questionnaaire (PHQ), Depression-Anxiety-Stress Scale-21 (DASS-21), Beck Depression Inventory (BDI), Hamilton depression rating scale (HAMD), etc were used for depression screening. Next, different categories for the severity of disease were reported in different studies. Moreover, even though in the same country, there were different diagnostic criteria applied for psychological distress studies. For example, in China, ten different diagnostic tools were used to detect depression compared with three diagnostic tools used in India, which give rise to a broad range of prevalence of depression in China (31.7%-41.2%) and India (27.9%-39.3%). In our meta-analysis, there was a wide gap in terms of sample size in all 148 studies, ranged from 100 to 14,825 in this review. All these factors would explain why there was a high heterogeneity for the prevalence of psychological distress, for instance, depression in this study.

Comment

9. Please explain the reason for publication bias in this study (in discussion section)

Answer: We have added the causes of high heterogeneity in the discussion session, last paragraph to read as:

Multiple factors contributed to publication bias, comprises of study design, sample size, decision of authors, journal editors and reviewers [187]. We had excluded all the unpublished data and studies with a sample size of below 100 in this review. However, we need to weigh the advantage and disadvantages as those published studies have gone through a rigorous review process, which gave a more reliable result compared with those unpublished data. Furthermore, we also excluded studies with outcomes that did not fulfil our operational definition of psychological distress. This systematic only included manuscripts wrote in English due to constraint of resources. Thus, we had to interpret the results of this systematic review carefully within the context of its limitations level and research setting [186].

---

## [Editor Report · Decision Letter 1]

15 Sep 2021

Psychological Distress Among Healthcare Providers During COVID-19 In Asia: Systematic Review and Meta-Analysis

PONE-D-21-21203R1

Dear Dr. Cheong,

We’re pleased to inform you that your manuscript has been judged scientifically suitable for publication and will be formally accepted for publication once it meets all outstanding technical requirements.

Kind regards,

Ali Rostami

Academic Editor

PLOS ONE

Additional Editor Comments (optional): The manuscript is acceptable, but references should be re-formatted according to journal's style.
---

## [Editor Report · Acceptance letter]

21 Sep 2021

PONE-D-21-21203R1 

Psychological Distress Among Healthcare Providers During COVID-19 In Asia: Systematic Review and Meta-Analysis 

Dear Dr. Cheong:

I'm pleased to inform you that your manuscript has been deemed suitable for publication in PLOS ONE. Congratulations! Your manuscript is now with our production department. 

Kind regards, 

on behalf of

Dr. Ali Rostami 

Academic Editor

PLOS ONE